# Mild Intermittent Cold Stimulation Affects Cardiac Substance Metabolism via the Neuroendocrine Pathway in Broilers

**DOI:** 10.3390/ani13223577

**Published:** 2023-11-19

**Authors:** Yuanyuan Liu, Lu Xing, Yong Zhang, Xiaotao Liu, Tingting Li, Shijie Zhang, Haidong Wei, Jianhong Li

**Affiliations:** 1College of Life Science, Northeast Agricultural University, Harbin 150030, China; lyy3648@163.com (Y.L.); xinglu9292@163.com (L.X.); zy1225735339@163.com (Y.Z.); wangpc1008@163.com (X.L.); liting18337257656@163.com (T.L.); zsj18054341215@163.com (S.Z.); 2Key Laboratory of Chicken Genetics and Breeding, Ministry of Agriculture and Rural Affairs, Harbin 150030, China

**Keywords:** cold adaptation, cold stress, neuroendocrine, lipid metabolism, glucose metabolism

## Abstract

**Simple Summary:**

In the present study, we investigated the effects of two intermittent cold stimulation conditions (3 °C and 9 °C below the normal brooding temperature) on the neuroendocrine and cardiac substance metabolism pathways in broilers. We analyzed the transcriptome and the expression levels of neuroendocrine substances in serum, the mRNA expression of cardiac substance metabolism-related genes and heat-shock proteins. The results of our study showed that mild cold stimulation (at 3 °C lower than the normal temperature) can activate the neuroendocrine pathway release neurotransmitters and improve cardiac substance metabolism to protect the broilers from cold stress damage.

**Abstract:**

This study aimed to investigate the impact of cold adaptation on the neuroendocrine and cardiac substance metabolism pathways in broilers. The broilers were divided into the control group (CC), cold adaptation group (C3), and cold-stressed group (C9), and experimental period was divided into the training period (d 1–35), recovery period (d 36–43), and cold stress period (d 43–44). During the training period, the CC group was reared at ambient temperature, while C3 and C9 groups were reared at 3 °C and 9 °C lower than the ambient temperature, respectively, for 5 h/d at 1 d intervals. During the recovery period, all the groups were maintained at 20 °C. Lastly, during the cold stress period, the groups were divided into two sub-groups, and each sub-group was placed at 10 °C for 12 h (Y12) or 24 h (Y24) for acute cold stimulation. The blood, hypothalamic, and cardiac tissues samples were obtained from all the groups during the training, recovery, and acute stress periods. The results revealed that the transcription of *calcium voltage-gated channel subunit alpha 1 C (CACNAIC)* was increased in the hypothalamic tissues of the C3 group (*p* < 0.05). Moreover, compared to the CC group, the serum norepinephrine (NE) was increased in the C9 group (*p* < 0.05), but insulin (INS) was decreased in the C9 group (*p* < 0.05). In addition, the transcription of the *phosphoinositide-3 kinase (PI3K)*, *protein kinase B (Akt)*, *mammalian target of rapamycin (mTOR)*, *SREBP1c*, *FASN*, *ACC1*, and *SCD* genes was down-regulated in the C3 and C9 groups (*p* < 0.05); however, their expression increased in the C3 and C9 groups after acute cold stimulation (*p* < 0.05). Compared to the CC group, the transcription of *forkhead box O1 (FoxO1)*, *PEPCK*, *G6Pase*, *GLUT1*, *HK1*, *PFK*, and *LDHB* genes was up-regulated in the C3 and C9 groups (*p* < 0.05. Furthermore, compared to the CC and C9 groups, the protein and mRNA expressions of *heat shock protein (HSP) 70* and *HSP90* were significantly increased in the C3 group (*p* < 0.05). These results indicate that intermittent cold training can enhance cold stress tolerance in broilers by regulating their neuroendocrine and cardiac substance metabolism pathways.

## 1. Introduction

The concept of “stress” was introduced by Selye in 1936, and it refers to the non-specific reactions of individuals after exposure to various stressors [1]. Cold stress occurs when the ambient temperature falls below the optimal temperature required for the normal functioning of an organism. It leads to an imbalance between heat production and heat dissipation in the body, as well as a series of deleterious physiological and functional responses [2]. Adaptation is the process by which cells and tissues adjust their metabolism, function, and structure in response to changes in the internal and external environments, and enhance their tolerance to similar stimuli in the future [3]. Studies found that repeated cold stimulation can lead to cold adaptation in animals. During cold adaptation, the temperature feedback control system regulates the endocrine and metabolic processes to allow the body to establish a new equilibrium, thereby reducing cold stress damage [4,5]. Su et al. reported that broilers housed at 3 °C lower than the control broilers had structurally intact ileum and reduced expression of pro-inflammatory cytokines after experiencing acute cold stimulation at 7 °C [6]. Therefore, cold adaptation can improve an organism’s ability to resist cold and maintain normal physiology.

The calcium voltage-gated channel subunit alpha 1 C (CACNA1C) is the main channel for Ca^2+^ influx into cardiomyocytes, and it is involved in a variety of physiological processes, such as cardiac contraction, neurotransmitter release, and gene expression regulation [7,8]. Increased serum epinephrine (E) and norepinephrine (NE) concentrations regulate energy metabolism and transformation by activating the mammalian target of rapamycin (mTOR) and forkhead box O1 (FoxO1) signaling pathways [9,10,11]. The phosphoinositide-3 kinase (PI3K)/protein kinase B (Akt)/mTOR signaling pathway directly regulates the activity of sterol regulatory element-binding protein 1c (SREBP1c), which further regulates the expression of adipose synthesis-related genes, including acetyl-CoA carboxylase 1 (ACC1), fatty acid (FA) synthase (FASN), and stearoyl-CoA desaturase (SCD), which increase energy storage in the form of lipids [12,13]. Moreover, upon activation, the PI3K/Akt pathway promotes the inactivation of FoxO1, thereby decreasing its transcriptional activity. This results in a decrease in the expression of glucose-6-phosphatase (G6Pase) and phosphoenolpyruvate carboxy kinase (PEPCK), which are involved in glycogen metabolism and gluconeogenesis, leading to a decrease in blood glucose levels [14]. Glucose transporter 1 (GLUT1) facilitates the transfer of glucose across the cell membrane of mammalian cells, even during cold stress, thereby providing essential energy for the body [15]. Hexokinase1 (HK1), the initial irreversible rate-limiting enzyme in glycolysis, catalyzes the formation of glucose-6-phosphate (G6P) and the phosphorylation of other hexoses, while phosphofructokinase (PFK), a critical regulatory enzyme in the glycolytic pathway, catalyzes the conversion of fructose-6-phosphate (F-6-P) to fructose-1,6-bisphosphate (F-1,6-P) [16,17]. Lactate dehydrogenase (LDH) B, a glycolytic enzyme, has been identified as a potential marker for evaluating environmental stress, and exposure to low temperatures increases the expression of LDHB, thus promoting the conversion of pyruvate to lactic acid, thereby promoting energy metabolism [18]. Insulin (INS) is a neuroendocrine hormone that stimulates the synthesis and conversion of fats and regulates energy storage [19]. Adiponectin (ADPN) is an insulin-sensitizing hormone that plays a role in maintaining blood sugar and lipid levels by promoting gluconeogenesis and FA oxidation, respectively [20,21]. Zhang et al. demonstrated that broilers exposed to low temperatures developed cold tolerance by increasing ADPN expression [22].

Heat shock proteins (HSPs) protect organisms from various types of stresses and serve as indicators of injury in the animal stress response system [23,24]. For instance, HSP70 aids in the proper folding of nascent peptide chains and the repair of misfolded proteins under various stresses, thereby increasing stress tolerance in animals [25].

Cardiac metabolism is another important factor in stress response, and its regulation is necessary for adapting to a variety of physiological conditions and meeting specific energy requirements [26,27]. Wei et al. suggested that severe cold stimulation at 9 °C below normal rearing temperature induces cardiomyocyte inflammation and apoptosis via Nrf2/HO-1 pathway-related oxidative stress in broilers, and that mild cold stimulation can improve cardiac adaptability to a cold environment [28]. However, the mechanism by which intermittent cold training influences cardiac metabolism in broilers is unclear. Therefore, this study aimed to investigate the effects of two different intermittent cold stimulation conditions on the neuroendocrine and cardiac metabolism pathways in broilers.

## 2. Materials and Methods

### 2.1. Animal Experimental Procedure

Two hundred and eighty-eight one-day-old broilers were randomly divided into 3 groups and housed in different artificial climate houses. The broilers were raised under proper humidity (1–14 d, kept at 60–70%; 15–43 d, kept at 40–50%), an illumination system (1–3 d, light for 24 h; 4–43 d, light for 23 h), illumination intensity (1–14 d, 40 Lux; 15–43 d, 15 Lux), NH_3_ concentration (1–21 d, less than 5 ppm; 22–43 d, less than 10 ppm), and CO_2_ concentration (less than 800 ppm), and were allowed free access to food and water. All chicks are immunized in strict accordance with broiler production standards. All animal procedures in this study were approved by the Northeast Agricultural University Animal Care and Use Committee (Application No. IACUCNEAU20150616).

### 2.2. Experimental Design

On days 1–3, two hundred and eighty-eight one-day-old broilers were kept in a 35 °C environment; on days 4–14, the rearing temperature was lowered by 0.5 °C per day; and on days 15–32, the broilers were randomly divided into 3 groups (control group: CC; cold adaptation group: C3; and cold stress group: C9) with 6 replicates of 16 broilers in each group. The broiler rearing temperature in the CC group continued to drop by 0.5 °C every day, and broilers in the C3 and C9 groups were placed in an environment that was 3 and 9 °C lower than that of the CC group, respectively, for 5 h on alternate days (15:00–20:00). On days 33–35, the broiler rearing temperature in the CC group was maintained at 20 °C, and broilers in the C3 and C9 groups continued to be placed in an environment that was 3 and 9 °C lower than that of the CC group, respectively, on alternate days. On days 36–43, all groups of broilers were maintained at 20 °C. On day 44, all groups of broilers were placed in a 10 °C environment for 12 h (Y12 group) and 24 h (Y24 group) for acute cold stimulation, respectively. The specific cold training temperature program is shown in Table 1 and Figure 1.

### 2.3. Sample Collection and Processing

On the 22nd, 29th, 36th, 43rd, and 44th day, samples of blood were collected and placed at 4 °C, then centrifuged at 3000 rpm for 15 min, and the supernatant was extracted for ELISA detection. The cardiac and hypothalamus tissues of broilers were collected, washed with 0.9% sodium chloride and immediately stored in liquid nitrogen, then long-term stored in a freezer at −80 °C for RT-qPCR and Western Blot detection.

### 2.4. Total RNA Extraction and Complementary DNA (cDNA) Library Construction

Total RNA from broiler hypothalami was extracted with TRIzol reagent (Invitgen, Carlsad, CA, USA), and RNA integrity was assessed using a Bioanalyzer 2100 RNA Nano 6000 Assay Kit. Next, mRNA was enriched from total RNA using magnetic beads containing Dyna Oligo (dT), and the mRNA was fragmented into 200–300 bp fragments using fragmentation buffer and heat treatment. The first cDNA strand was synthesized using mRNA fragments as templates, as well as random oligonucleotides and SuperScrip II, and the second cDNA strand was synthesized by replacing. After that, the cDNA ends were repaired, tailed, and ligated to the indexing connector. Lastly, the final cDNA library was selected and enriched for appropriate fragments by PCR, and the cDNA library was analyzed using the Illumina HiSeq 2000 system.

### 2.5. Transcriptome Sequencing and Bioinformatics Analysis

Raw reads from RNA-seq were filtered using Cutadapt (1.16) software to obtain high-quality data by removing reads with junctions and average quality scores below Q20. The filtered reads were then compared to a reference genome using HISAT2, an upgraded version of TopHat2 (2.1.1) (http://ccb.jhu.edu/software/hisat2/index.shtml (accessed on 17 December 2020)) software. Using the DESeq (1.38.3) R software package, DEGs were screened based on the multiplicity of differences |log2FoldChange| > 1 and a significant *p*-value < 0.05. The DEGs were subsequently made into bar graphs and were plotted as volcano maps using the R language ggplots2 (3.4.1) software package. Differential genes were analyzed in combination with all comparison group samples using the R language Pheatmap (1.0.12) software package. The Euclidean method was used for calculating distances, and the hierarchical clustering longest distance method (Complete Linkage) was used for clustering. GO enrichment analysis was performed using topGO, and the *p*-value was calculated using the hypergeometric distribution method (the criterion for significant enrichment is a *p*-value of <0.05). At the end, KEGG enrichment analysis was performed using clusterprofiler. In the analysis, the list of genes and the number of genes in each pathway was calculated using the KEGG pathway-annotated differential genes, and then the *p*-value was calculated using the hypergeometric distribution method (the criterion for significant enrichment was *p*-value < 0.05).

### 2.6. Detection of Blood Biochemical Indexes

The E, NE, INS, and ADPN content were evaluated using ELISA kits (Xinle, Wuhan, China) according to the manufacturer’s instructions. Each sample was measured in triplicate, and its average value was calculated as the final value. The optical density (OD) value was then measured at the corresponding wavelength of each index using a miniature flat panel reader (BiotekInstrument Inc., Winooski, VT, USA). Finally, the concentration of each index of each sample was calculated according to the corresponding standard curve and OD value.

### 2.7. Quantitative Real-Time PCR (RT-qPCR)

Total RNA was extracted from cardiac and hypothalamus samples using TRIzol, and cDNA was synthesized by reverse transcription kit (Toyobo, Japan). PCR was performed using a THUNDERBIRD^®^ SYBR^®^ qPCR Mix reagent kit (Toyobo, Japan) on a AriaMx Real-Time PCR machine (Agilent, Santa Clara, CA, USA) according to the manufacturer’s instructions. β-actin was used as the internal reference gene, and the relative expression of the target gene was calculated according to the 2^−∆∆CT^ method. Primer sequences are shown in Appendix A.

### 2.8. Western Blot

The total proteins from cardiac tissue homogenates was extracted using NP-40 lysis buffer (SparkJade, Shandong, China) supplemented with 1% phenyl methane sulfonyl fluoride (PMSF) (SparkJade, Shandong, China) or with 0.5% PMSF and 0.5% phosphatase inhibitors (SparkJade, Shandong, China). The bicinchoninic acid (BCA) protein quantification method was used to ensure that the concentration of each sample was basically equal. Then, the equal amounts of protein lysates were separated using 10% sodium dodecyl sulfate polyacrylamide gel electrophoresis (SDS-PAGE), and the separated proteins on the gel were transferred to a polyvinylidene difluoride (PVDF) membrane. The PVDF membranes were blocked with 5% (*w*/*v*) skim milk powder for 2 h and incubated at 4 °C overnight with the following primary antibodies HSP70 (Wanleibio, Shenyang, China, 1:700), HSP90 (Wanleibio, Shenyang, China, 1:900), β-actin (Wanleibio, Shenyang, China, 1:8000). Secondary antibodies, HRP-labeled Goat Anti-Rabbit IgG (H+L) (ABclonal, Wuhan, China, 1:10,000), were used to incubate membranes at room temperature for 1 h. Target proteins were visualized using an enhanced chemiluminescence kit (MILIBO, Massachusetts, USA). Images were captured using the Gel-ProAnalyzer. And each band was analyzed using ImageJ (1.8.0.345) software (Bio-Rad Laboratories, Hercules, CA, USA), and the quantification of expression levels was based on expression ratios of target protein/β-actin.

### 2.9. Statistical Analysis

Analysis was performed using IBM SPSS Statistics 21.0 (IBM, Armonk, NY, USA), Kolmogorov–Smirnov test was carried out on the data, and the data that did not obey the normal distribution showed homogeneity of variance after data transformation. According to experimental design, serum index content, protein, and mRNA relative expression data were statistically analyzed using a multivariate analysis of variance (ANOVA) using a general linear model. The data are presented as mean ± standard deviation (mean ± SD), with significant differences for *p* < 0.05.

## 3. Results

### 3.1. Analysis of Hypothalamic Transcriptome Data of Intermittent Cold Training of Broilers

#### 3.1.1. Transcriptome Sequencing Data Comparison

The RNA-seq and mapping results of intermittent cold training broilers are shown in Table 2 and Figure 2. As can be seen from Table 2, the total number of sequences used for Clean Reads is between 3.6 × 10^7^ and 4.4 × 10^7^, the total number of bases (Clean Base) is between 6.05 × 10^9^ and 7.00 × 10^9^, and the percentage of bases with base recognition accuracy greater than 99.9% (Q30) is between 90% and 91%. Figure 2A shows the results of correlation analysis between samples. The correlation coefficient is more than 0.90, indicating the reliability of surveying and mapping data. Figure 2B shows the results of principal component analysis (PCA) of samples between groups, which shows that there are significant differences in principal components between groups. Therefore, the quality of the sample sequencing data is good and can be used for follow-up analysis.

#### 3.1.2. Analysis and Verification of Differential Expression Genes in Hypothalamus

There were 100 differentially expressed genes (Figure 3A,B), including 52 up-regulated genes and 48 down-regulated genes. The clustering analysis of DEGs showed that six samples from two groups had obvious clustering and high-and low-expression gene distribution between groups (Figure 3C). Then, the differential analysis of gene expression by DESeq, and the selected differential genes were annotated to GO. The genes and their products were described and classified based on three aspects: molecular function, cell module function, and biological process. As shown in Figure 3D, in the biological process, cellular response to organonitrogen, response to external stimulus, and response to stimulus enrich most DEGs. In addition, in molecular function, DEGs are mainly enriched through proximal promoter sequence-specific DNA binding and RNA polymerase II transcription factor-binding entries. Then, the signal pathway of DEG enrichment was found by KEGG pathway annotation, and the biological function of DEGs in the species was obtained. In this experiment, the KEGG results are plotted as a bubble diagram (Figure 3E), and 20 pathways are shown. The pathways of enriching DEGs are influenza A, phagosome, neuroactive ligand-receptor interaction, adrenergic signaling in cardiomyocytes, and so on.

In addition, Figure 3F showed that the expressions of key genes cluster of differentiation 36 (CD36), calcium voltage-gated channel subunit alpha1C (CACNA1C), the tryptophan hydroxylase2 (TPH2) and sestrin1 (SESN1) in PPARα signal pathway, folate biosynthesis, and the adrenergic and p53 signal pathway were remarkably up-regulated, while those of epidermal growth factor receptor (EGFR), nuclear receptor subfamily 4 group A member 1 (NR4A1), TNF superfamily member 10 (TNFSF10) and aldo-keto reductase family 1 member D1 (AKR1D1) in primary bile acid biosynthesis, mitogen-activated protein kinase (MAPK), and FoxO signal pathway were clearly down-regulated. These results were verified by RT-qPCR technology and found to be consistent with the transcriptome results (Figure 3G).

### 3.2. Effects of Intermittent Cold Training on Blood Biochemical Indices of Broilers

As shown in Figure 4, the interaction between Group (CC, C3, and C9) and Time (22 d, 29 d and 36 d) exhibited a significant effect on serum E, NE, ADPN, and INS (*p* < 0.05) in the broilers. The concentration of E in the serum of broilers was not significantly influenced by Group (*p* > 0.05) and Time (*p* > 0.05). The concentrations of NE and INS were influenced by Group (*p* < 0.05) and Time (*p* < 0.05). While the concentration of NE in the CC group was significantly lower than the C9 group (*p* < 0.05), its concentration was not different between the C3 and CC or C9 groups (*p* > 0.05). Moreover, the concentration of NE was significantly increased with Time (*p* < 0.05). Meanwhile, the concentration of INS in the C9 group was significantly lower than in the CC and C3 groups (*p* < 0.05), but its concentration was not different between the C3 and CC groups (*p* > 0.05). The concentration of ADPN was not significantly influenced by Group (*p* > 0.05) but was influenced by Time (*p* < 0.05).

### 3.3. Effects of Intermittent Cold Training on PI3K/Akt/mTOR Signaling Pathway and Its Downstream Key Genes in the Cardiac Muscle of Broilers

As shown in Figure 5, the interaction between Group and Time exhibited a significant effect on the mRNA expression levels of *mTOR*, *SREBP1c*, *SCD*, *ACC1,* and *FASN* (*p* < 0.05) but no significant effect on the mRNA expression levels of *PI3K* and *Akt* (*p* < 0.05). The mRNA expression level of *PI3K* was not significantly influenced by Group (*p* > 0.05) but was significantly increased with Time (*p* < 0.05). The mRNA expression levels of *Akt*, *mTOR,* and *FASN* in the C3 and C9 groups were significantly lower than the CC group (*p* < 0.05), but its expression level was not different between C3 and C9 groups (*p* > 0.05). In addition, the expression level of *Akt* increased markedly with Time (*p* < 0.05). The mRNA expression levels of *SREBP1c* and *SCD* in the C9 group were significantly lower than of the CC and C3 groups (*p* < 0.05), but its expression level was not different between the CC and C3 groups (*p* > 0.05). And the expression level of *SREBP1c* on the 36 d was lower than 29 d (*p* < 0.05). Moreover, the mRNA expression level of *SCD* was significantly decreased with Time (*p* < 0.05). The mRNA expression level of *ACC1* in the CC group was significantly higher than the C3 and C9 groups (*p* < 0.05), but its expression level was not different between C3 and C9 groups (*p* > 0.05).

### 3.4. Effects of Intermittent cold Training on FoxO1 Signaling Pathway and Its Downstream Key Genes in the Cardiac Muscle of Broilers

As shown in Figure 6, the interaction between Group and Time exhibited a significant effect on the mRNA expression levels of *LDHB*, *PEPCK*, *G6Pase*, *PFK,* and *HK1* (*p* < 0.05), but the mRNA expression levels of *FoxO1* and *GLUT1* were not different between Group and Time (*p* > 0.05). And the mRNA expression levels of *LDHB*, *GLUT1*, *G6Pase*, *PFK,* and *HK1* were significantly influenced by Group and Time (*p* < 0.05), the mRNA expression levels of *FoxO1* and *PEPCK* were significantly influenced by Time (*p* < 0.05). The mRNA expression level of *LDHB* in the CC group were significantly lower than of the C3 and C9 groups (*p* < 0.05), but its expression level was not different between the C3 and C9 groups (*p* > 0.05). And the expression level of *LDHB* on the 36 d was higher than 22 d (*p* < 0.05). Additionally, the mRNA expression level of *GLUT1* in the C3 group was significantly higher than the C9 group (*p* < 0.05), and the mRNA expression level of *GLUT1* on the 36 d was significantly higher than the 22 d and 29 d (*p* < 0.05). The mRNA expression level of *GLUT1* in the C3 group was significantly higher than the CC group (*p* < 0.05). Meanwhile, the mRNA expression level of *PFK* in the C3 group was significantly higher than the CC and C9 groups (*p* < 0.05), but its expression level was not different between CC and C9 groups (*p* > 0.05). The mRNA expression level of *PFK* at 36 d was significantly lower than at 29 d (*p* < 0.05). Lastly, the mRNA expression level of *HK1* at 36 d was significantly lower than for the 22 d and 29 d groups (*p* < 0.05), but its expression level was not different between the 22 d and 29 d groups (*p* > 0.05).

### 3.5. Effects of Acute Cold Stress on HSPs and Blood Biochemical Indexes in the Cardiac Muscle of Broilers

As shown in Figure 7, the interaction between Group (Y0, Y12, and Y24) and Time exhibited a significant effect on the mRNA expression levels of *HSP70* and *HSP90* (*p* < 0.05) and the concentrations of E, NE, ADPN and INS (*p* < 0.05), but the protein expression levels of HSP70 and HSP90 were not different between Group and Time (*p* > 0.05). And the mRNA expression levels of *HSP90*, and the concentrations of E, NE, and INS were significantly influenced by Group and Time (*p* < 0.05), the protein expression level of HSP70 was significantly influenced by Time (*p* < 0.05), and the concentration of ADPN was significantly influenced by Group (*p* < 0.05). However, the protein expression level of HSP90 and the mRNA expression level of *HSP70* were not significantly influenced by Group and Time (*p* > 0.05). The protein expression level of HSP70 in the C3 group was significantly higher than in the CC and C9 groups (*p* < 0.05), but its expression level was not different between the CC and C9 groups (*p* > 0.05). Furthermore, the mRNA expression level of *HSP90* in the CC group was significantly lower than the C3 and C9 groups (*p* < 0.05). The concentration of E in the C9 group was significantly higher than the CC group (*p* < 0.05). The concentrations of NE and INS in the CC group was significantly lower than in the C3 and C9 groups (*p* < 0.05), but their concentrations were not different between the C3 and C9 groups (*p* > 0.05). Moreover, the concentration of INS in the Y0 group was significantly higher than in the Y12 and Y24 groups (*p* < 0.05), but its concentration was not different between the Y12 and Y24 groups (*p* > 0.05).

### 3.6. Effects of Acute Cold Stress on PI3K/Akt/mTOR Signaling Pathway and Its Downstream Key Gene mRNA Expression Levels in the Cardiac Muscle of Broilers

There were significant interactions between Group and Time (Figure 8) affecting the mRNA expression levels (*p* < 0.05). The mRNA expression levels of *Akt*, *mTOR*, *SREBP1c*, *ACC1,* and *FASN* were significantly influenced by Group and Time (*p* < 0.05), but the mRNA expression levels of *PI3K* and *SCD* were not significantly influenced by Time (*p* > 0.05). The mRNA expression level of *PI3K* in the Y24 group was higher than in the Y12 group (*p* < 0.05). The mRNA expression level of *Akt* in the CC group was significantly lower than in the C3 and C9 groups (*p* < 0.05), but its expression level was not different between the C3 and C9 groups (*p* > 0.05). Meanwhile, the mRNA expression level of *Akt* in the Y24 group was higher than in the Y0 group (*p* < 0.05). Furthermore, the mRNA expression level of *SREBP1c* in the C9 group was significantly higher than in the CC group (*p* < 0.05). The mRNA expression levels of *SCD* and *FASN* were not significantly influenced by Group (*p* > 0.05), however, its mRNA expression level in the Y12 was higher than Y0 and Y24 (*p* < 0.05). The mRNA expression level of *ACC1* in the CC group was significantly lower than the C3 and C9 groups (*p* < 0.05), but its expression level was not different between C3 and C9 groups (*p* > 0.05). Additionally, the expression level of *FASN* increased markedly with Time (*p* < 0.05).

### 3.7. Effects of Acute Cold Stress on FoxO1 Signaling Pathway and Its Downstream Key Gene mRNA Expression Levels in the Cardiac Muscle of Broilers

There were significant interactions between Group and Time (Figure 9) affecting the mRNA expression levels (*p* < 0.05). The mRNA expression levels of *FoxO1*, *LDHB*, *PEPCK*, *GLUT1*, *G6Pase*, *PFK,* and *HK1* were significantly influenced by Group and Time (*p* < 0.05). The mRNA expression level of *FoxO1* in the Y12 group was significantly lower than in the Y0 and Y24 groups (*p* < 0.05), but its expression level was not different between the Y0 and Y24 groups (*p* > 0.05). Moreover, the mRNA expression level of *LDHB* in the C3 group was significantly higher than in the CC and C9 groups (*p* < 0.05), but its expression level was not different between the CC and C9 groups (*p* > 0.05). While the mRNA expression level of *LDHB* in the Y24 group was significantly lower than in the Y0 and Y12 groups (*p* < 0.05), its expression level was not different between the Y0 and Y12 groups (*p* > 0.05). However, the mRNA expression level of *PEPCK* was not significantly influenced by Group and Time (*p* > 0.05). The mRNA expression level of *GLUT1* was not different between the groups (*p* > 0.05). Furthermore, the mRNA expression level of *GLUT1* in the Y0 group was significantly lower than in the Y12 and Y24 groups (*p* < 0.05), but its expression level was not different between the Y12 and Y24 groups (*p* > 0.05). Meanwhile, the mRNA expression level of *G6Pase* in the CC group was significantly lower than the C3 and C9 groups (*p* < 0.05), but its expression level was not significantly influenced by the Time (*p* > 0.05). The mRNA expression level of *PFK* in the C3 group was significantly higher than in the CC group (*p* < 0.05), and that in the Y12 group was significantly higher than in the Y0 group (*p* < 0.05). In addition, the mRNA expression level of *HK1* in the C9 group was significantly higher than in the CC group (*p* < 0.05), but its expression level was not significantly influenced by Time (*p* > 0.05).

## 4. Discussion

Cold stress is a significant factor affecting animal health and welfare and has a detrimental impact on the economic performance of the livestock and poultry industries. Previous research findings from our group indicated that broilers can achieve cold adaptation following a period of cold training, accompanied by improved immune function and enhanced cold stress resistance [5]. In the present study, we investigated the effect of two intermittent cold stimulation conditions: 3 °C and 9 °C below the normal rearing temperature of broilers. Transcriptome and KEGG pathway enrichment analyses revealed that the MAPK, cardiac myocyte adrenergic, FoxO, and neuroactive ligand–receptor interaction pathways were enriched in the C3 group. Moreover, the expression of *CACNA1C* was up-regulated in the C3 group. CACNA1C regulates calcium ion inflow and participates in a variety of physiological processes, including cardiac contraction, neurotransmitter release, and gene expression regulation [7,8]. Additionally, CACNA1C stimulates the secretion of NE and E, which induce lipolysis and thermogenesis to meet the body’s increased energy demands during cold stress [29]. In this study, the concentration of NE was significantly increased with time, suggesting that increased NE secretion increases heat production in broilers, thereby improving their cold stress tolerance and self-preservation ability. In addition, the serum E concentration decreased in the C9 group during the training period, indicating that severe cold stress may impair adrenocortical function and disrupt the endocrine system. After the training and recovery periods, the three groups were subjected to acute cold stress for 12 and 24 h and their response was evaluated. The serum E concentration of the C3 group was increased, indicating that appropriate cold training during early development can improve cold stress tolerance in broilers.

Previous studies have reported that the PI3K/Akt/mTOR pathway can regulate adipogenesis by modulating SREBP1c, a key transcription factor that controls lipid metabolism [30]. Being a trans-endoplasmic reticulum membrane protein, SREBP1c detects a wide range of metabolic signals and is enabled by specific protein hydrolysates, which cleave and release SREBP1c [31,32]. SREBP1c then translocates to the nucleus, where it stabilizes the transcription of its target genes, such as ACC1, FASN, and SCD, which are involved in the adipogenesis pathway [33]. In this study, the transcription of *PI3K/Akt/mTOR* and its downstream genes was significantly reduced in the C9 group, suggesting that intense cold conditions can alleviate cold stress injury by inhibiting lipid synthesis and decreasing adenosine triphosphate (ATP) consumption. The transcription of *SCD* increased in the C3 group, indicating that cold adaptation can activate *SCD* to induce unsaturated FA synthesis, improve membrane fluidity, and enhance cold stress tolerance. After acute cold stress for 12 and 24 h, the transcription of *SCD* and *ACC1* increased significantly in the C9 group, indicating an increase in FA synthesis. ADPN stimulates FA oxidation and inhibits lipid synthesis. Under cold stress, the expression of ADPN increases to promote lipolysis for energy production and consumption [34,35]. In this study, the ADPN concentration in the C3 group initially increased and then decreased prior to acute cold stress, whereas it increased continuously in the C9 group. These findings suggest that the cardiac metabolism of the C3 group could promote energy production to resist cold stress and prevent cold stress injury. INS has an opposing effect to that of ADPN, and it promotes lipid synthesis and regulates serum amino acid and FA concentrations [36]. Additionally, the serum INS concentration decreases with an increase in cold stress intensity. In this study, the serum INS concentration in the C3 and C9 group decreased significantly. Furthermore, the transcription of *HSP70* and *HSP90* was significantly increased in the cardiac tissues of the C3 group after acute cold stress, indicating high cold stress resistance in this group.

Glucose metabolism is an essential metabolic process that provides energy and regulates various physiological processes [37]. FoxO1, a member of the fork box protein family, plays a pivotal role in regulating fundamental cellular processes, such as cell differentiation, glucose metabolism, and cell cycle arrest [38,39,40]. Moreover, FoxO1 can up-regulate the expression of PEPCK and G6Pase, which are key enzymes involved in gluconeogenesis, thus ultimately promoting insulin resistance [41]. In this study, the transcription of *FoxO1* and its downstream targets, including *PEPCK* and *G6Pase*, were increased in both the C3 and C9 groups, suggesting a potential role of FoxO1 signaling pathway in regulating blood glucose homeostasis under cold stress. Furthermore, the expression of *GLUT1*, a member of the glucose transporter family, decreased continuously in the cardiac tissues of the C9 group, indicating an inhibition in glucose transport in vivo [42]. HK catalyzes the conversion of glucose to G6P, and further regulates other metabolic pathways, such as the pentose phosphate pathway to produce synthetic metabolic intermediates; therefore, variations in HK expression levels reflect glucose utilization [43]. PFK catalyzes the phosphorylation of F-6-P to F-1,6-P and has a significant impact on glucose consumption and energy production [44]. In this study, the transcription of *HK1* and *PFK* in the cardiac tissues of the C3 and C9 groups was up-regulated, indicating accelerated glycolysis in broilers under cold stress. However, the transcription of *HK1* and *PFK* returned to normal expression levels in the C3 group with an increase, their transcription remained up-regulated in the C9 group, likely due to cold stress-induced metabolic disturbance and glycolysis inhibition. LDH is composed of LDHA and LDHB and plays a critical role in the regulation of energy metabolism in the cardiac and muscle tissues [45,46]. In the cardiac tissues of broilers, LDHB utilizes nicotinamide adenine dinucleotide and NADH as cofactors to catalyze the conversion of pyruvate and lactic acid in the final step of glycolysis [47]. Pyruvate is the substrate for ATP production in mitochondria; therefore, the expression of LDHB may be directly associated with mitochondrial function and energy production. In this study, the transcription of *LDHB* was significantly increased in the C3 and C9 groups, suggesting an increase in glycolysis in broilers under cold stress. Furthermore, following acute cold stress, the transcription of most of the glucose metabolism-related genes did not change significantly in the cardiac tissues of the C3 group, indicating that cold adaptation is beneficial for mitigating cold stress damage.

## 5. Conclusions

The findings of this study showed that mild cold stimulation (at 3 °C lower than the ambient temperature) increased cold stress tolerance in broilers by inducing the release of neurotransmitters and activating the PI3K/Akt/mTOR and FoxO1 signaling pathways to increase their cardiac metabolism. But it was the opposite in severe cold stimulation at 9 °C below normal rearing temperature. Therefore, cold adaptation improved cardiac metabolism via the neuroendocrine pathway to promote cold stress tolerance in broilers.

## Figures and Tables

**Figure 1 animals-13-03577-f001:**
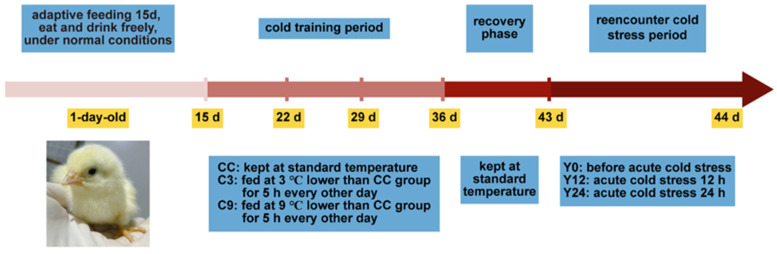
The cold training temperature scheme.

**Figure 2 animals-13-03577-f002:**
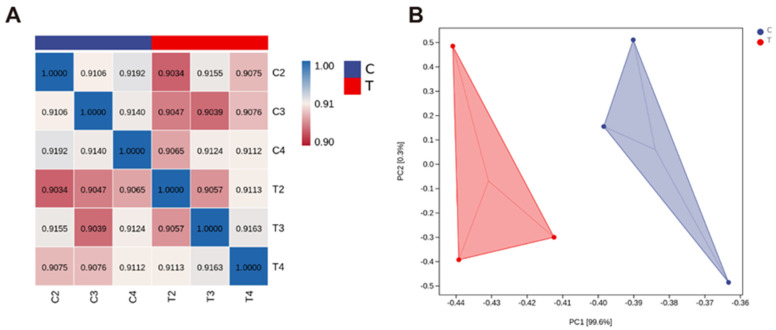
Results of (**A**) correlation analysis and (**B**) PCA of control group (C) and cold adaptation group (T) hypothalamic samples by RNA-seq analysis in broilers. Data are presented as mean ± standard (n = 3).

**Figure 3 animals-13-03577-f003:**
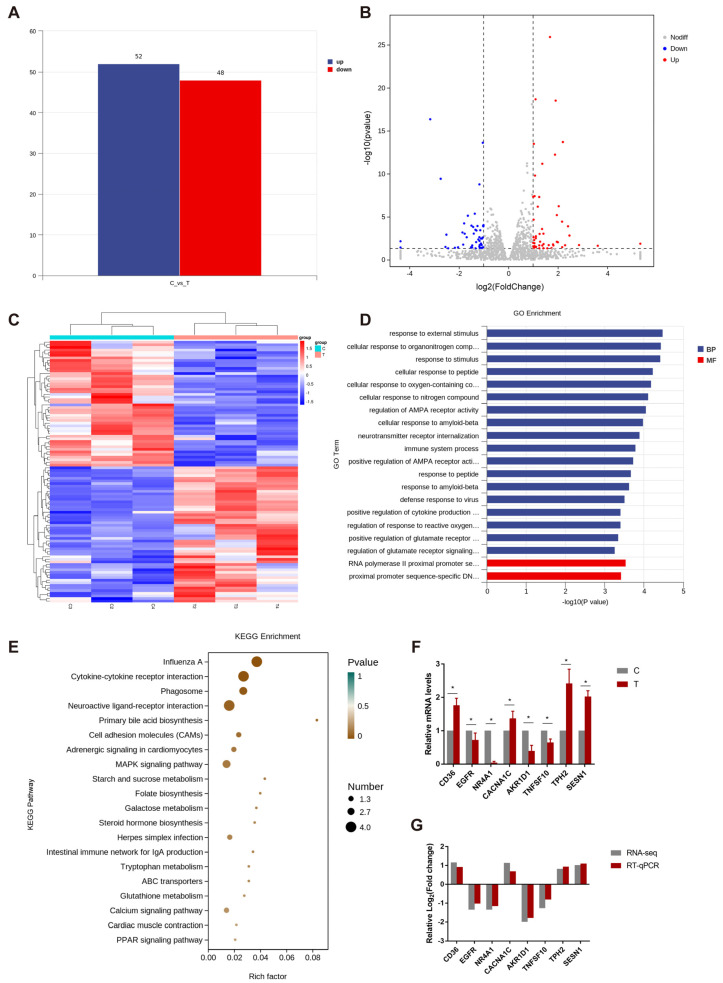
Results of the analysis and verification of differentially expressed genes in hypothalamus. (**A**) Histogram of the number of differentially expressed genes, (**B**) volcano plot, and (**C**) clustering map of DEGs between hypothalamic samples of C and T groups by RNA-seq analysis in broilers. (**D**) GO and (**E**) KEGG enrichment analysis of DEGs between hypothalamic samples of C and T groups by RNA-seq analysis in broilers. (**F**) Relative mRNA expression of eight screened genes in C and T group was detected by RT-qPCR. (**G**) Comparison of expression levels of candidate genes as per RNA-seq and RT-qPCR analysis. Data are presented as mean ± standard (n = 3). * *p* < 0.05.

**Figure 4 animals-13-03577-f004:**
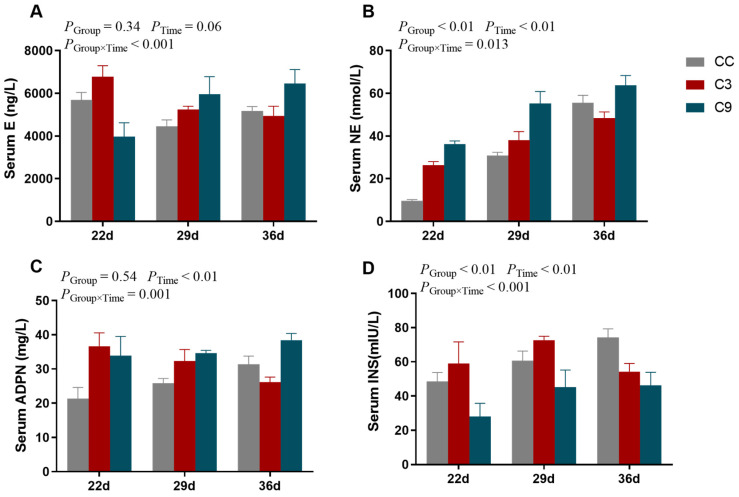
Results of ELISA detection of blood biochemical indexes of broilers after intermittent cold training. The content of (**A**) E, (**B**) NE, (**C**) ADPN, and (**D**) INS in different groups (control group: CC; cold adaptation group: C3; and cold stress group: C9) and days old (22 d, 29 d, and 36 d). Data are presented as mean ± standard (n = 3).

**Figure 5 animals-13-03577-f005:**
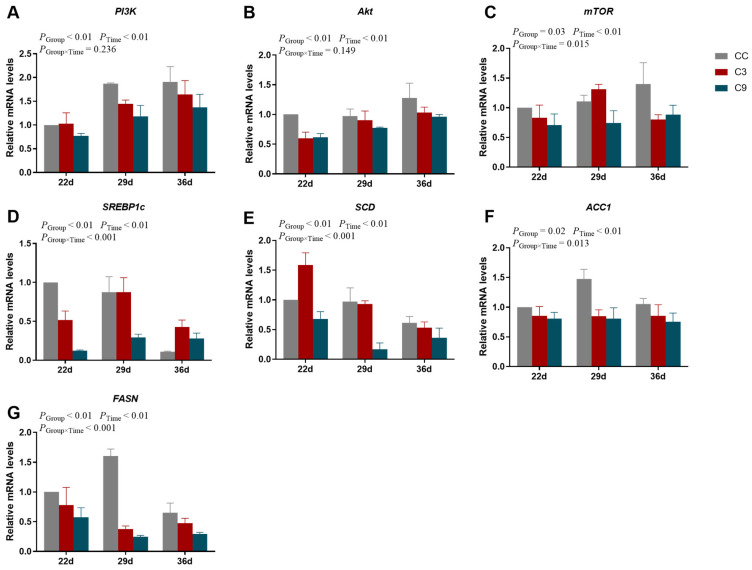
Results of intermittent cold training on PI3K/Akt/mTOR signaling pathway in the cardiac muscle of broilers. The mRNA expression levels of (**A**) *PI3K*, (**B**) *Akt*, (**C**) *mTOR*, (**D**) *SREBP1c*, (**E**) *SCD*, (**F**) *ACC1,* and (**G**) *FASN* in different groups (CC, C3, and C9) and days old (22 d, 29 d, and 36 d). Data are presented as mean ± standard (n = 4).

**Figure 6 animals-13-03577-f006:**
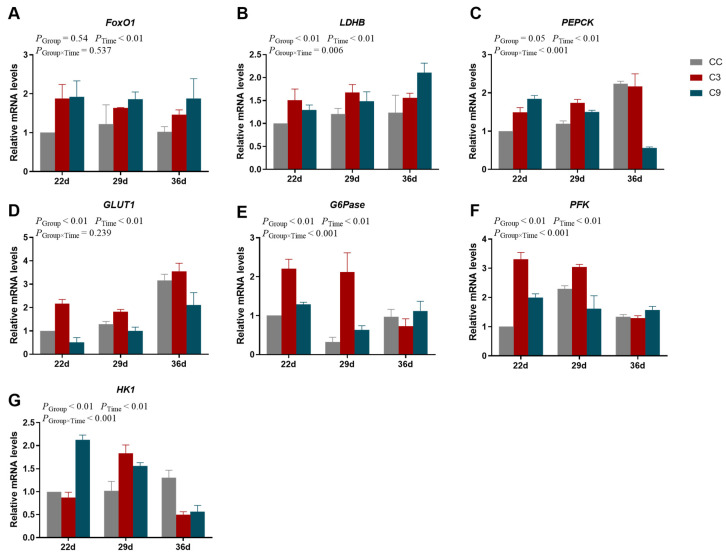
Results of intermittent cold training on FoxO1 signaling pathway in the cardiac muscle of broilers. The mRNA expression levels of (**A**) *FoxO1*, (**B**) *LDHB*, (**C**) *PEPCK*, (**D**) *GLUT1*, (**E**) *G6Pase*, (**F**) *PFK,* and (**G**) *HK1* in different groups (CC, C3, and C9) and days old (22 d, 29 d, and 36 d). Data are presented as mean ± standard (n = 4).

**Figure 7 animals-13-03577-f007:**
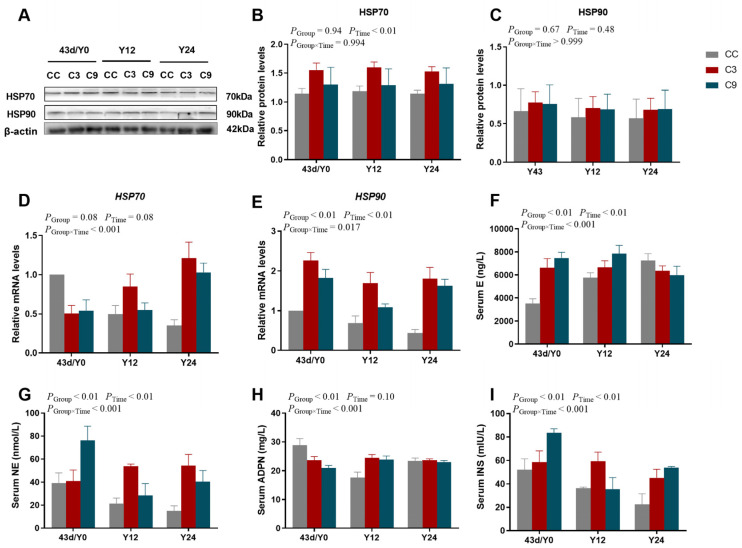
Results of acute cold stress on HSPs and blood biochemical indexes in the cardiac muscle of broilers. (**A**) HSP70, HSP90, and β-actin protein expression levels in each group. (**B**,**C**) Quantification of expression levels based on expression ratios of HSP70/β-actin and HSP90/β-actin. Data are presented as mean ± standard (n = 3). (**D**) *HSP70* and (**E**) *HSP90* in different groups (CC, C3, and C9) and duration of acute cold stress (Y0, Y12, and Y24). Data are presented as mean ± standard (n = 4). The content of (**F**) E, (**G**) NE, (**H**) ADPN, and (**I**) INS in each group. Data are presented as mean ± standard (n = 3).

**Figure 8 animals-13-03577-f008:**
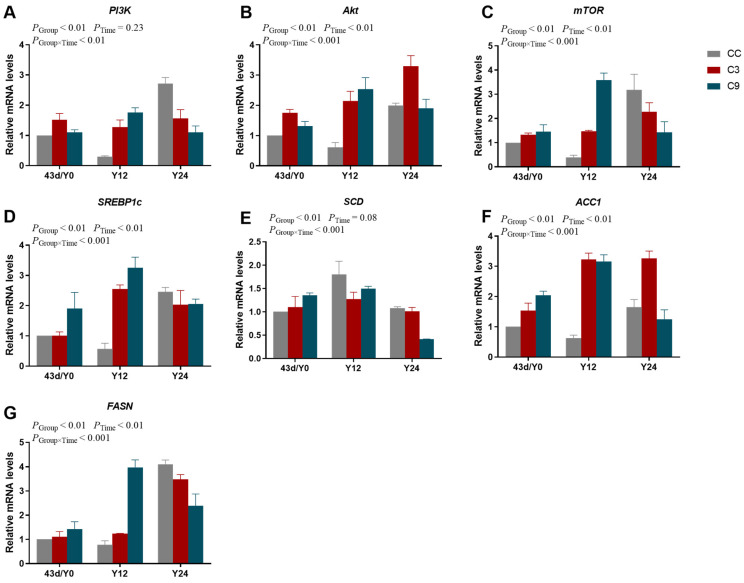
Results of acute cold stress on PI3K/Akt/mTOR signaling pathway in the cardiac muscle of broilers. The mRNA expression levels of (**A**) *PI3K*, (**B**) *Akt*, (**C**) *mTOR*, (**D**) *SREBP1c*, (**E**) *SCD*, (**F**) *ACC1,* and (**G**) *FASN* in different groups (CC, C3, and C9) and duration of acute cold stress (Y0, Y12, and Y24). Data are presented as mean ± standard (n = 4).

**Figure 9 animals-13-03577-f009:**
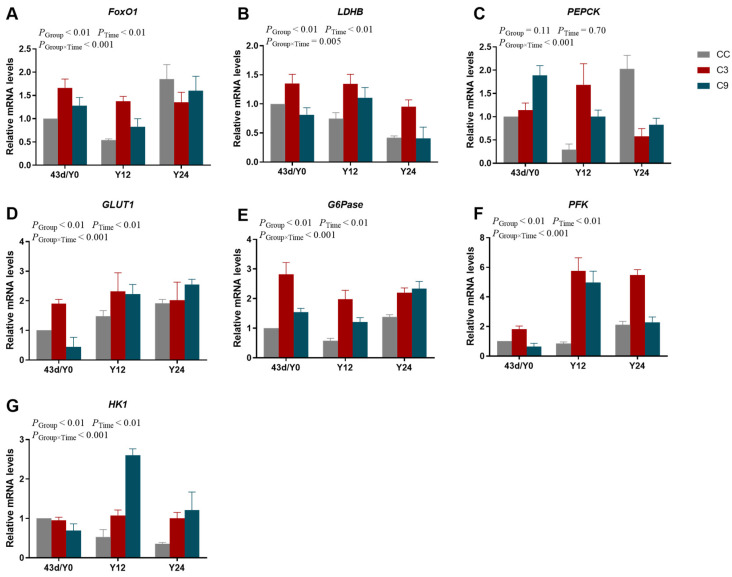
Results of acute cold stress on FoxO1 signaling pathway in the cardiac muscle of broilers. The mRNA expression levels of (**A**) *FoxO1*, (**B**) *LDHB*, (**C**) *PEPCK*, (**D**) *GLUT1*, (**E**) *G6Pase*, (**F**) *PFK,* and (**G**) *HK1* in different groups (CC, C3, and C9) and duration of acute cold stress (Y0, Y12, and Y24). Data are presented as mean ± standard (n = 4).

**Table 1 animals-13-03577-t001:** Temperature regimens of cold adaptation.

Days Old	Control Group (CC)	Cold Adaptation Group (C3)	Cold Stress Group (C9)
1–3	35 °C		
4–14	reduce by 0.5 °C per day		
15	29	26	20
16	28.5	28.5	28.5
17	28	25	19
18	27.5	27.5	27.5
19	27	24	18
20	26.5	26.5	26.5
21	26	23	17
22	25.5	25.5	25.5
23	25	22	16
24	24.5	24.5	24.5
25	24	21	15
26	23.5	23.5	23.5
27	23	23	23
28	22.5	19.5	13.5
29	22	22	22
30	21.5	18.5	12.5
31	21	18	12
32	20.5	20.5	20.5
33	20	17	11
34	20	20	20
35	20	17	11
36–43	20		
44	10	10	10

**Table 2 animals-13-03577-t002:** Statistical table of hypothalamic sequencing data.

Samples	Clean Reads	Clean Base (%)	Q30 Proportion (%)	Map Reads	Map Reads (100%)
C1	37,662,272	6,050,529,300	90.13	34,104,512	90.55%
C2	39,519,606	6,358,805,100	90.05	35,815,992	90.63%
C3	40,606,524	6,517,980,600	90.19	36,767,361	90.55%
C4	40,374,598	6,476,575,200	90.20	36,754,473	91.03%
T1	41,284,360	6,643,700,100	90.86	37,607,189	91.09%
T2	43,470,024	6,999,076,500	90.38	39,304,084	90.42%
T3	41,271,848	6,644,863,800	90.03	37,481,197	90.82%
T4	38,195,746	6,149,473,800	90.57	34,858,213	91.26%

## Data Availability

Data are contained within the article.

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
