# Peer review of "Mild Intermittent Cold Stimulation Affects Cardiac Substance Metabolism via the Neuroendocrine Pathway in Broilers"

_animals, 2023, doi:10.3390/ani13223577_

Round 1

Reviewer 1 Report (New Reviewer)

Comments and Suggestions for Authors

The authors present interesting data on intermittent cold training affects cardiac substance metabolism through neuroendocrine pathway in broilers. The manuscript is generally well written. However, there are some issues need to be addressed:

(1) What is the relationship between hypothalamic transcriptome sequencing and cardiac substance metabolism

(2) Lines 21-22 lacks an introduction to temperature treatment.

(3) Line 25, the full name of CACNAIC.

(4) Lines 26-30 is a discussion statement, it is not appropriate to put it here, it is recommended to delete it.

(5) Lines 35-43, where the results of the C9 group are missing.

(6) The writing format of line 76 "day" is not consistent with that of line 171 “d”. Please unify the full text.

(7) Introduction is too long. Please refine the introduction.

(8) Experimental designHow is the cold adaptation scheme determined?

(9) Western blotDetail the secondary antibodies and the concentrations that have been used.

(10) Line 146 of the article uses the term "cold exposure". What's the difference between cold exposure and cold stress? please agree in the article.

(11) Line 293, there is a space in the middle of 90%, please delete it. Check the full text format.

Author Response

Reviewer 2 Report (New Reviewer)

Comments and Suggestions for Authors

There are some large issues with this manuscript and the way that the data was analyzed and presented.  

I am not sure why the authors chose to analyze the transcriptome data the way they did.  It has been divided into control and experimental groups (two groups), meanwhile the authors have three groups with different sampling times.  So I don't feel that they way it is presented provides any meaningful information.  

According to materials and methods the authors did a one way anova for statistical analysis of Western blot, and mRNA expression data.  But that is not appropriate because they more than two variables, so a multi-variate analysis should be carried out.  

My advice would be to basically redo all the statistical analysis, and provide the transcriptome analysis for all the experimental groups.

The other big issue here is the English language.  The summary has to be completely re-written as it makes no sense.  Throughout the manuscript that authors use long sentences that are hard to understand and follow.  The introduction is way too long, and it includes background research from pigs, chickens etc....but it just jumps around.  Maybe just concentrate on poultry research?

The authors also need to remember that protein and gene expression doesn't necessarily correlate with function, so that should be taken into consideration when writing the discussion and results.

Minor comments

Italicize gene names

57 what is heat to meat ratio

64 Alpine regions?

86 Buidling should be developing

95 adaptation change to adapt

102 who is they?

120 as a change to is a

123 protein inositol kinase

130-137 doesn't make sense

170-170 last sentence in the introduction.  I don't think that the authors are revealing any mechanisms in this study.

174 Two hundred and eighty eight, one day old broilers.

203 Delate 8 am.  Just say that the samples were taken at 12 and 24 hours post stress.

205 Centrifuged not centrifugated.

208 freezer not fridge

216 by substitution

In some places of the material and methods the authors describe everything in almost too much detail, while in others not enough.

Stop using the word additionally all the time.

236-238 incomprehensible sentence

240 delete thereafter

Section 2.5 this section needs to be simplified.

260 was calculated

277 skim milk not skimmed

was the secondary antibody labeled? how were the blots visualized? was a chromogenic substance used? add information.

Throughout the manuscript the authors jump between past and present tense.  Please make it consistent.

Comments on the Quality of English Language

This needs to be improved.  The sentences are frequently too long, hard to understand.  In many cases the sentences do not make sense, and strange word combinations are used.  I recommend the authors possible paying an editorial service to improve the language.  I understand that it is difficult, but a manuscript as written must be presented in a way that can be understood.

Round 2

Reviewer 2 Report (New Reviewer)

Comments and Suggestions for Authors

The authors need to address the comments I raised point by point.  They should upload another document describing the revisions they did.

I have gone over the manuscript and I can tell that many of the comments I submitted were not addressed.  

Comments on the Quality of English Language

Still needs improvement.

There is an error in the first sentence- breeding temperature, should be brooding temperature.  

I carefully went over the English verbiage and grammar for the first revision, now it is up to the authors to correctly implement them.

Author Response

Dear reviewers and editors,

We wish to thank you for the time and efforts you have spent reviewing our paper. Those comments are all valuable and very helpful for revising and improving our paper. We have studied these comments carefully and modified our manuscript accordingly. The changes have been highlighted in red throughout the manuscript. We hope the modifications have addressed all the shortcomings outlined. In particular, this revised manuscript of our resubmitted has been significantly improved as follows.

Sincerely yours Jianhong Li

The authors need to address the comments I raised point by point. They should upload another document describing the revisions they did. I have gone over the manuscript and I can tell that many of the comments I submitted were not addressed.

Answer: I apologize to the teacher for any confusion I may have caused due to the way I responded, as I did make major changes to the article as per the teacher's comments, including simple summary, abstract, introduction, materials and methods, results, figures, discussion and conclusions. I will reorganize my revised response and mark the article in red, please point out anything wrong, thank you very much for your comments.

Question: Comments on the Quality of English Language. Still needs improvement.

Answer: Thank you for your valuable suggestions. We have asked a professional team to polish the language of the article, corrected the grammar and the ambiguous sentences.

Question: There is an error in the first sentence- breeding temperature, should be brooding temperature.

Answer: Thank you for your valuable advice. We have revised “breeding temperature” to “brooding temperature” (Line 11, Page 1).

Question: I carefully went over the English verbiage and grammar for the first revision, now it is up to the authors to correctly implement them.

Answer: Thank you for your professional advice. We have revised the article according to the first revision, and here is the response to the first revision.

First revision

Comments and Suggestions for Authors

There are some large issues with this manuscript and the way that the data was analyzed and presented.

Answer: First of all, thank the expert for your affirmation of our work. And those comments are all valuable and very helpful for revising and improving our paper. According to your nice suggestions, we have made extensive corrections to our previous draft, the detailed corrections are listed below.

Question 1: I am not sure why the authors chose to analyze the transcriptome data the way they did. It has been divided into control and experimental groups (two groups), meanwhile the authors have three groups with different sampling times. So I don't feel that they way it is presented provides any meaningful information.

Answer: Thank you for your valuable question. First of all, this experiment is to explore the impact of mild intermittent cold stimulation on broilers, thus we have chosen to analyze control and experimental groups (3°C below the normal brooding temperature). But this adaptive stimulus sometimes has no significant impact on the body compared with the control group. Therefore, according to previous research results (doi.org/10.3390/ani12233260, doi.org/10.1016/j.jtherbio.2023.103658), we added the cold stress group in the follow-up experiment, which can more highlight the impact of cold adaptation on broilers. In addition, we chose the 36th day of cold acclimatization maturation for transcriptome sequencing of broilers, and based on the results, QRT-PCR and serum biochemical assay experiments were conducted for other time points, which was to observe the metabolic changes in broilers during the establishment of cold acclimatization. Finally, we subjected the broilers to acute cold stimulation again, which was to observe the metabolic changes of broilers after acclimatizing to the cold environment and then encountering cold stimulation.

Question 2: According to materials and methods the authors did a one way anova for statistical analysis of Western blot, and mRNA expression data.  But that is not appropriate because they more than two variables, so a multi-variate analysis should be carried out. My advice would be to basically redo all the statistical analysis, and provide the transcriptome analysis for all the experimental groups.

Answer: Thank you for your valuable suggestions. We have carried out multiple analyses on all the data in the paper according to your advice, reanalyzed the results as well as graphing (Lines 262-407, Pages 9-14). Accordingly, we have modified the description of results section of the abstract and discussion. However, due to the long storage time of the samples, transcriptome sequencing could not be carried out, so the transcriptome data of anther experimental group could not be analyzed. Please forgive me.

Question 3: The other big issue here is the English language. The summary has to be completely re-written as it makes no sense. Throughout the manuscript that authors use long sentences that are hard to understand and follow. The introduction is way too long, and it includes background research from pigs, chickens etc....but it just jumps around. Maybe just concentrate on poultry research?

Answer: Thank you for your valuable advices. We have rewritten the summary, revised long sentences (Lines 10-17, Page 1). Moreover, we have rewritten the introduction and deleted inappropriate content (Lines 44-58, Page 2).

Simple Summary: In the present study, we investigated the effects of two intermittent cold stimulation conditions (3°C and 9°C below the normal breeding temperature) on the neuroen-docrine and cardiac substance metabolism pathways in broilers. We analyzed transcriptome and the levels of neuroendocrine substances in serum, the mRNA levels of cardiac substance metab-olism-related genes and heat shock proteins. The results of our study showed that mild cold stimulation (at 3°C lower than the normal temperature) can activate the neuroendocrine pathway release neurotransmitters and improve cardiac substance metabolism to protect the broilers from cold stress damage.

Introduction: The concept of “stress” was introduced by Selye in 1936, and it refers to the non-specific reactions of individuals after exposure to various stressors [1]. Cold stress occurs when the ambient temperature falls below the optimal temperature required for the normal functioning of an organism. It leads to an imbalance between heat produc-tion and heat dissipation in the body, as well as a series of deleterious physiological and functional responses [2]. Adaptation is the process by which cells and tissues adjust their metabolism, function and structure in response to changes in the internal and external environments and enhances their tolerance to similar stimuli in the future [3]. Studies found that repeated cold stimulation can lead to cold adaptation in animals. During cold adaptation, the temperature feedback control system regulates the endo-crine and metabolic processes, to allow the body to establish a new equilibrium, thereby reducing cold stress damage [4,5]. Su et al. reported that broilers housed at 3°C lower than the control broilers had structurally intact ileum and reduced levels of pro-inflammatory cytokines after experiencing acute cold stimulation at 7°C [6]. Therefore, cold adaptation can improve an organism’s ability to resist cold and maintain normal physiology.

Question 4: The authors also need to remember that protein and gene expression doesn't necessarily correlate with function, so that should be taken into consideration when writing the discussion and results.

Answer: Thank you for your valuable suggestions. We have rewritten the discussion and results, and chosen words more carefully (Lines 408-494, Pages 14-16).

Minor comments

Question 5: Italicize gene names

Answer: Thank you for pointing out our mistake. We have revised and checked the full paper.

Question 6: 57 what is heat to meat ratio

Answer: Thank you for your correction. The “heat to meat ratio” should be revised to “feed to meat ratio”, the feed to meat ratio refers to the amount of feed consumed by raising livestock and poultry to gain one kilogram. It is an important index to evaluate feed reward.

Question 7: 64 Alpine regions?

Answer: Thank you for pointing out our mistake. We have changed “Alpine regions” to “High latitude region”.

Question 8: 86 Buidling should be developing

Answer: Thank you for pointing this out. This is due to my careless expression. We have revised “Building cold adaptation therefore improves an animal's ability to resist cold as well as maintain normal physiology” to “During cold adaptation, the temperature feedback control system regulates the endocrine and metabolic processes, to allow the body to establish a new equilibrium, thereby reducing cold stress damage” (Lines 52-55, Page 2).

Question 9: 95 adaptation change to adapt

Answer: Thank you for pointing this out. We have revised “adaptation” to “adapt”.

Question 10: 102 who is they?

Answer: Thanks for your professional suggestions and sorry for making you confused. We have changed “Heat shock proteins (HSPs) are an indicator of injury in the animal stress response system [19]. They protect organisms from various types of stress by increasing the ex-pression of HSPs, thereby improving their tolerance to stress [20].” to “Heat shock proteins (HSPs) protect organisms from various types of stresses and serve as indicators of injury in the animal stress response system” and they are Heat shock proteins (HSPs) (Lines 89-90, Page 2).

Question 11: 120 as a change to is a

Answer: Thank you for pointing this out. We have rewritten the introduction and deleted this sentence.

Question 12: 123 protein inositol kinase

Answer: We were really sorry for our careless mistakes. We have revised “phosphorus inostitide-3-kinase (PI3K)” to “phosphoinositide-3 kinase (PI3K)” (Lines 31-32, Page 1) (Lines 64-65, Page 2).

Question 13: 130-137 doesn't make sense

Answer: Thank you for your valuable suggestions. We have deleted this section because it doesn't fit here.

Question 14: 170-170 last sentence in the introduction. I don't think that the authors are revealing any mechanisms in this study.

Answer: Thank you for pointing this out. We have changed “This would reveal the underlying mechanisms of intermittent cold training on cardiac substance metabolism in broilers.” to “Therefore, this study aimed to investigate the effects of two different intermittent cold stimulation conditions on the neuroendocrine and cardiac metabolism pathways in broilers.” (Lines 100-102, Page 3).

Question 15: 174 Two hundred and eighty eight, one day old broilers.

Answer: Thank you for your correction. We have changed “288 one-day-old healthy broilers” to “two hundred and eighty-eight, one day old broilers” (Line 115, Page 3).

Question 16: 203 Delate 8 am. Just say that the samples were taken at 12 and 24 hours post stress.

Answer: Thank you for your valuable suggestions. We have changed “At 8:00 in the morning on the 22nd, 29th, 36th and 43rd day, and on the 44th day (12 h and 24 h after acute cold stress),” to “On the 22nd, 29th, 36th, 43rd and 44th day, the samples of…” (Lines 133-135, Page 4).

Question 17: 205 Centrifuged not centrifugated.

Answer: Thank you for pointing this out. We've modified “centrifugated” to “Centrifuged” (Line 134, Page 4).

Question 18: 208 freezer not fridge

Answer: Thank you for your correction. We have revised “refrigerator” to “freezer” (Line 137, Page 4).

Question 19: 216 by substitution

Answer: Thank you for pointing this out. We have revised “by substitution” to “by replacing” (Line 145, Page 4).

Question 20: In some places of the material and methods the authors describe everything in almost too much detail, while in others not enough.

Answer: Thanks for your valuable suggestions. We have shortened section 2.5 (Lines 150-167, Pages 4-5) and supplemented section 2.8 (Lines 182-198, Page 5) according to your advice.

Question 21: Stop using the word additionally all the time.

Answer: Thanks for your nice suggestions. According to your advice, we have used “In addition, furthermore, moreover etc.” to substituted “additionally”.

Question 22: 236-238 incomprehensible sentence

Answer: Thank you for pointing this out. This is due to my careless expression. We have changed “GO enrichment analysis was performed using topGO, where the list of genes and the number of genes per term was calculated using the differential genes annotated by the GO terms, and the P-value was calculated using the hypergeometric distribution method (the criterion for significant enrichment is a P-value of < 0.05).” to “GO enrichment analysis was performed using topGO, and the P-value was calculated using the hypergeometric distribution method (the criterion for significant enrichment is a P-value of < 0.05).” (Lines 161-163, Page 5).

Question 23: 240 delete thereafter

Answer: Thank you for pointing this out. We have deleted “thereafter” (Lines 183-184, Page 5).

Question 24: Section 2.5 this section needs to be simplified.

Answer: Thank you for your valuable suggestion. We have simplified section 2.5 according to your advice “Raw reads from RNA-seq were filtered using Cutadapt software to obtain high quality data by removing reads with junctions and average quality scores below Q20. The filtered Reads were then compared to a reference genome using HISAT2, an up-graded version of the TopHat2 (http://ccb.jhu.edu/software/hisat2/index.shtml) soft-ware. Using the DESeq R software package, DEGs were screened based on the multi-plicity of differences |log2FoldChange| > 1 and a significant P-value < 0.05. The DEGs were subsequently made into bar graphs and DEGs were plotted as volcano maps using the R language ggplots2 software package. Differential genes were analyzed in combi-nation with all comparison group samples using the R language Pheatmap software package. The Euclidean method was used for calculating distances, and the hierarchical clustering longest distance method (Complete Linkage) was used for clustering. GO enrichment analysis was performed using topGO, and the P-value was calculated using the hypergeometric distribution method (the criterion for significant enrichment is a P-value of < 0.05). At the end, KEGG enrichment analysis was performed using clusterprofiler. In the analysis, the list of genes and the number of genes in each pathway were calculated using the KEGG pathway-annotated differential genes, and then the P-value was calculated using the hypergeometric distribution method (the criterion for significant enrichment was P-value < 0.05).” (Lines 150-167, Pages 4-5).

Question 25: 260 was calculated

Answer: Thank you for pointing this out. We have revised “were calculated” to “was calculated” (Line 165, Page 5).

Question 26: 277 skim milk not skimmed

Answer: Thank you for pointing this out. We have revised “skimmed milk” to “skim milk” (Line 191, Page 5).

Question 27: was the secondary antibody labeled? how were the blots visualized? was a chromogenic substance used? add information.

Answer: Thank you for your correction. We have added these information “Secondary antibodies HRP-labeled Goat Anti-Rabbit IgG (H+L) (ABclonal, China, 1:10000) were used to incubate membranes at room temperature for 1 h. Target proteins were visualized with an enhanced chemiluminescence kit (MILIBO, USA).” (Lines 193-196, Page 5).

Question 28: Throughout the manuscript the authors jump between past and present tense.  Please make it consistent.

Answer: We were really sorry for our careless mistakes. We have revised “present tense” to “past tense” and checked the full article.

Question 29: Comments on the Quality of English Language

This needs to be improved. The sentences are frequently too long, hard to understand. In many cases the sentences do not make sense, and strange word combinations are used. I recommend the authors possible paying an editorial service to improve the language. I understand that it is difficult, but a manuscript as written must be presented in a way that can be understood.

Answer: Thank you for your valuable suggestions. We have asked a professional team to polish the language of the article, corrected the grammar and the ambiguous sentences.

This manuscript is a resubmission of an earlier submission. The following is a list of the peer review reports and author responses from that submission.

Round 1

Reviewer 1 Report

Comments and Suggestions for Authors

Dear Authors, 

Although this study seems to be highly interesting and important, I see multiple contradictions. In the material and method section, wrote that the study included 288 broiler chickens divided into three groups with 3 repetitions in each, but in the hypothalamus transcriptome study, only 8 animals were sequenced and later only 6 were used to presentation of results. Why two samples were removed from the analysis during hypothalamus analysis was not explained. It is not clear which analyses were performed on which number of animals it should be clarified, including gene expression in the heart of broilers,  in western blot analysis and blood parameters. Moreover, for western blot analysis, the criteria for selection proteins are not indicated. In the methods was informed that the study used the Cufflinks tool for DEG analysis, and in the results section that Deseq was used. In the abstract, the abbreviations are not clarified. Moreover, I do not see the relationships between heart qPCR analysis, blood parameters and hypothalamic transcriptome. The title concerns hypothalamic transcriptome and the discussion almost the whole is about cardiac differences during cold adaptation. This all should be rearranged. I encourage you to resubmit after these corrections.

Reviewer 2 Report

Comments and Suggestions for Authors

This study is designed to investigate the impact of cold adaptation on cardiac substance metabolism in broilers, as well as its underlying mechanism. The findings from this study provide some valuable information, which will help improve animal production under cold condition. However, I have the following concerns that need to be addressed before consideration for publication.

1. Title: the original title could not well cover all the research content. This study performed analysis on hypothalamus, blood and heart. Just based on hypothalamic transcriptome analysis, the study cannot reveal that cold training affects cardiac substance metabolism in broilers. I think the authors should revise the title. I suggest “Intermittent cold training affects cardiac substance metabolism through neuroendocrine pathway” for reference.

2. Simple summary: this summary seems to have multiple objectives. Please revise it to make it clear to readers.

3. Abstract: the full names of the abbrevations including SAM, E, NE, INS and ADPN should be spelled out in the abstract.

4. Abstract: CACN1C should be CACNA1C, activation should refer to the activation of protein, however, this study did not present any data in regard to the activation of CACNA1C protein. This gene was just metioned once in the Results section, not in the Discussion section. I wonder why this gene was mentioned in the abstract as its importance was not introduced in the manuscript.

5.  Abstract: Line 42, it looks better to replace the creation of a cold adaptation model with cold adaptation.

6. Introduction: I think the significance of cold adaptation in animal production should be introduced more.

7. Introduction: Line 54-56, please provide some references.

8. Introduction: Line 70, remove Namely.

9. Introduction: Line 76-108, this paragraph has litte information closely related to cold adaptation, except for Lines 95 and 106. Moreover, the authors should give some introudction in regard to the relationship between the signaling pathways (the PI3K/Akt/mTOR signaling pathway, the FOXO1 signaling pathway) and neuroendocrine hormones (E, NE, INS and ADPN). Otherwise, it is difficult for readers to understand how cold adaptation influence cardiac substance metabolism through neuroendocrine pathway.

10. Introduction: the authors also should provide more information on the progress in the field of cold stress and cold adaptation,especially the underlying molecular mechanisms.

11. Introduction: the authors did not provide the rationale for studying the heart but not other tissues. What progress has been made to the cold adaptation of the heart?

12. M&M: Line 116, 1-day-old should be revised as One-day-old. Line 119-120, the numbers in NH3 and CO2 should be presented as subscripts.

13. M&M: there are any birds dead during the experimental period? There are representative body weights for each group of birds?

14. M&M: Line 128, it should be 16 broilers each replicate. Line 129, what is the standard temperature? Please specify.

15. M&M: Line 140, revise 15 min was centrifuged with 3000 r/min.

16. M&M: Line 155, what reference gene was used for gene mapping? Please provide accession number.

17. M&M: the description on the methods for transcriptome sequencing and analysis is simple, please provide more details. For example, the criteria for screening differentially expressed genes were not provided. What programs or softwares were used for GO and KEGG analyses? Please supplement it.

18. M&M: Line 173, -∆∆CT should be presented as superscript. Line 202-203, the same problem. Please correct it.

19. Results: the sample size should be indicated in figure notes if applicable. Some figure notes did not provide sufficient information.

20. Results: this study just provide immunoblots of HSPs. Most of results supporting activation of signaling pathways (the PI3K/Akt/mTOR signaling pathway, the FOXO1 signaling pathway) are based on mRNA expression but not protein expression or activation. However, protein expression or activation is critical for activation of signaling pathway. I hope the authors can provide more immunoblots in regard to activation of the related signaling pathways. If not possible, please revise their conclusions or statements to make it so solid.

21. The manuscipt needs some editing work for wording and grammar. For example, Line 150-152, the past tense should be used. 

Comments on the Quality of English Language

moderate editing of English language is required